# Secondary Treatment of Mandibular Bone Fracture Using Sagittal Split Osteotomy and Segmentation of the Mandible: A Case Report

**DOI:** 10.3390/reports6020027

**Published:** 2023-06-06

**Authors:** Paweł Piotr Grab, Aldona Chloupek, Jakub Nowocień, Maciej Jagielak, Dariusz Jurkiewicz

**Affiliations:** 1Clinical Department of Cranio-Maxillo-Facial Surgery, Military Institute of Medicine—National Research Institute, Szaserów 128, 04-141 Warsaw, Poland; achloupek@wim.mil.pl (A.C.); jnowocien@wim.mil.pl (J.N.); mjagielak@wim.mil.pl (M.J.); 2Private Health Entity Ortognatyka, Aleja Krakowska 54, 02-256 Warsaw, Poland; 3Clinical Department of Otolaryngology, Military Institute of Medicine—National Research Institute, Szaserów 128, 04-141 Warsaw, Poland; djurkiewicz@wim.mil.pl

**Keywords:** 3D printing, sagittal split osteotomy, mandible fracture, secondary reconstructive surgery, case report

## Abstract

The secondary treatment of mandibular bone fractures poses a great challenge due to the complexity of several factors, such as incorrect primary fracture repositioning, inadequate internal fixation, nonunion, necrosis, local inflammation and infection, tooth loss, and malocclusion, serving as obstacles encountered by surgical teams. The aim of this case report is to detail the planning process, surgical technique, and outcome of the secondary treatment of the post-traumatic deformation, bone exposure, and partial necrosis of the mandible. The new approach described herein incorporated 3D planning and printing procedures, employing surgical techniques such as the segmentation of the mandible with unilateral sagittal split osteotomy and the vertical osteotomy of the mandibular body. New, stable occlusion; appropriate spatial relations; and proper osteosynthesis of the mandible were achieved. The encouraging results obtained demonstrate that the described method can be incorporated in similar cases of the secondary treatment of mandibular fractures and possibly lead to shorter hospitalization and convalescence and lower the associated costs.

## 1. Introduction

Mandibular bone fractures are the most common fractures of the viscerocranium that need surgical intervention [1,2,3]. The aim of the corresponding treatment is to restore pre-traumatic occlusion and facial aesthetics and thus correct jaw movement function. Initial interventions, both conservative and surgical, may not be successful due to the complexity of the relevant factors. For children; patients with preexisting malocclusion; partial or total edentulousness, and a poor condition of dentition; and polytraumatic, multidisciplinary cases, it can be a challenging to restore appropriate occlusion and correct the positioning of bone fragments. Inadequate fracture repositioning, reduction, and poor internal hardware fixation without proper stability can cause malocclusion, tooth loss, facial deformity, and bone necrosis, necessitating secondary treatment. Infection, nonunion, and poor function of the temporomandibular joint are additional factors [4].

Sagittal split osteotomy (SSO) is a common procedure used in maxillofacial surgery. It was first described by Trauner and Obwegeser in 1957. There are several modifications that have been introduced, such as the Dal Pont, Hunsuck, and Epcker variations, to enhance stability and convenience while reducing the complication rate.

The most common complications irrespective of the type of modification are unfavorable fracture lines, neurovascular bundle damage, infection, bleeding, and unstable osteosynthesis [5,6,7].

The primary use of SSO is orthognathic surgery. It is applicable in mandibular advancements, setbacks, ramus elongation, and all types of angular movements. However, it can be a great tool in the secondary treatment of the mandible fractures, e.g., the treatment of malocclusion resulting from condylar fractures with the shortening of the mandibular ramus [8,9] and reconstruction procedures of the mandible [10,11,12].

Virtual planning and 3D printing greatly impact oral and maxillofacial surgery. Patient-specific implants, splints, and cutting guides lead to higher precision, shorter surgery times, and more predictable outcomes. Trauma, reconstructive, and orthognathic surgery are a few of the treatment examples improved by this emerging technology. Improvements in 3D-printing and planning software have enabled the in-house production of high-quality surgical splints and guides, thus reducing the time and cost of preparation for surgery [13].

## 2. Case Report

### 2.1. Medical History

The patient was a 22-year-old man who was involved in a traffic accident at the age of 20, resulting in multi-organ injuries, corresponding to a Glasgow coma scale (GCS) of 5 points, consisting of peritoneal bleeding, second-degree spleen rupture, left and right femoral fractures, a multi-fragment fracture of the left humerus, and fractures of the left patella and left scaphoid bone. The patient’s cranial trauma consisted of intracerebral hematomas, the fracturing of all walls of the left maxillary sinus, a lower orbital edge fracture, impact fractures of both condyles, and a multi-fragment fracture of the mandibular body with the loss of teeth 44, 45, and 46.

The patient underwent general surgery and orthopedic interventions immediately under general anesthesia. Hemostasis in the craniofacial region was achieved. 

Primary reconstruction was performed on the 5th day of hospitalization. Internal fixation was performed using Tigerstaedt splints, yielding moderate results due to the partial loss of dentition and multifragment fractures. During the procedure, bone fragments of the left maxilla and orbit were repositioned and internally fixed using microplates and screws. The bony fragments of the left lower alveolar process were removed, bone parts were repositioned, and a rigid osteosynthesis using a reconstructive plate was performed on the mandible.

The patient was treated in the intensive care unit (ICU) department of the hospital. He was discharged after 3 weeks with follow-up recommendations. Unfortunately, the patient did not appear until 6 months after the primary surgery. 

The patient presented to our department in March 2022, 6 months after the primary treatment, with gum loss and septic inflammation with a purulent exudate in the mandibular fixation region. The bone surface and the reconstruction plate were partially exposed. Partial necrosis of the alveolar bone was also observed. A control orthopantomogram (OPG) and a computed tomography (CT) scan of the viscerocranium presented centripetal roll rotation of the right mandibular segment, a lack of the bone formation at the osteosynthesis site, and insufficient stabilization of the fracture (Figure 1). Furthermore, we observed malocclusion and difficulties in proper masticatory function on the right side of the patient’s dentition due to their partial lack of teeth and improper inward tilting of the lower right molars. 

### 2.2. Prior to Secondary Procedure Treatment

Parts of the necrotic bone were removed, and a swab was taken. After obtaining the results of antibiotic sensitivity testing conducted at the wound site, the patient was treated with targeted oral antibiotic therapy (clindamycin 300 mg 3 × 1; Clindamycin MIP 300^®^, MIP Pharma, Warsaw, Poland) for 14 days. During the follow-up visit, there was no inflammation, and the exposure site was clean (Figure 2B). Another facial CT scan was performed, and dental impressions were taken for the production of plaster models (Figure 2A). The patient was conscious, and no findings were noted in the oral cavity and nasopharynx. There were no palpable nodules in the patient’s head and neck area. The patient’s lab data and electrocardiogram showed no abnormalities. There was no history of surgical procedures prior to the trauma. Oral and written consent was obtained.

### 2.3. Presurgical Planning

Plaster models of dentition were CT-scanned with non-shading material placed between the upper and lower model. Digital imaging and communications in medicine format (DICOMs) of the facial CT and plaster model CT scans were imported into IPS Case Designer^®^ (KLS Martin Group, Tuttlingen, Germany), a virtual surgical planning software program. Three-dimensional reconstructions of both were combined in the program, and an accurate surgical model of the viscerocranium was exported as an STL file and three-dimensionally printed using a Next Dent 5100 printer (Next Dent, Soesterberg, The Netherlands) with Next Dent SG printing material (Next Dent, Soesterberg, The Netherlands).

The segmentation of the mandible using SSO and vertical osteotomy along with the removal of the necrotic mandibular bone was planned (Figure 3A–C). New occlusion with forward movement of the mandibular segment containing teeth no. 46, 47, and 48 was visualized by employing the plaster models and using an articulator. Subsequently, surgical acrylic splints (Meliodent Cold, HerausKulzer Ltd., Newbury, UK) were manually fabricated in the articulator. 

The 3D-printed mandible was cut according to the plan. A new position of the mandibular segment was established using an acrylic surgical splint, while the excess bone in the distal, necrotic site was simultaneously excised. 

Furthermore, a 3D reconstruction of the viscerocranium was transferred in the previously acquired STL format from IPS Case Designer^®^ into the Blender^®^ 3D program (Blender Foundation, Amsterdam, The Netherlands). After measuring the 3D model in vivo in the newly obtained position, surgical cutting guides were planned accordingly in the software and 3D-printed for the precise excision of the mandibular bone during the surgical procedure (Figure 3D–F).

### 2.4. Surgical Procedure

The procedure was divided into four steps. The first step consisted of grafting cancellous bone from the anterolateral surface of the left tibial bone 2 cm below the tuberosity. Using piezosurgery, a 1.5 cm × 1.5 cm square bone block was temporarily removed, revealing cancellous bone matter, which was collected using a bone spoon with a volume of approximately 3 cm^3^.

The second step began with the removal of the primary osteosynthesis material. The neurovascular bundle of the mandible on the right side was damaged during primary trauma. There was no visible bone union on site. Cutting guides were applied and stabilized in place on both sides of the excision with 2.0 titanium screws. Altered and necrotic bone, due to chronic inflammation and lack of stabilization, was removed with preplanned cutout margins using piezosurgery (Figure 4A–C). 

The third step involved performing an SSO according to the Dal Pont method on the right side of the mandible. Cortical cuts of the upper, anterior, and inferior bone borders were achieved via piezosurgery. The osteotomy line was cut in a curved line from the mandibular foramen to the level of the midline of tooth 46. This procedure was performed in this manner to achieve as much bony overlapping as possible. The split was completed using straight chisels. Full mobilization of the mandibular segment containing teeth No. 46, 47, and 48 was achieved (Figure 4D). The right inferior alveolar nerve was not preserved as it was entirely damaged during primary trauma.

The final step included placing intermaxillary fixation (IMF) screws (KLS Martin Group, Tuttlingen, Germany), five of which were inserted in the maxilla and four in the mandible. The acrylic splint was fitted on the upper dental arch. Mandibular segments were adapted based on the splint. The new occlusion and position of the mandibular parts were stabilized using metal wires and elastics. Furthermore, osteosynthesis of the mandible was performed using two 2.5 reconstruction titanium plates and screws (KLS Martin Group, Tuttlingen, Germany) (Figure 4E). The splint and IMF screws were removed, and the new occlusion was checked and approved by the surgical team. The scarred and altered parts of the mucosa at the primary reconstruction site were excised. Augmentation of the proximal part of the split and the contact side of the bone at the level of teeth 46 and 42 was performed using grafted cancellous bone (Figure 4F). The procedure was completed with double-layer wound closure. There were no complications during the surgery and the hospitalization period.

### 2.5. The Outcomes

The immediate results were satisfactory. Occlusion was stable; the passive mandible’s mobility was correct, with a 3.5 cm mouth opening; and wound closure was tension-free (Figure 5C).

The patient was not placed in intermaxillary fixation postoperatively nor at any time during the healing process. He was discharged on the third day after the procedure in a good general condition with limited local swelling at a level that was adequate for the procedure. The patient was recommended to adhere to a liquid diet for 14 days and a soft, semi-solid diet for the next 28 days and to avoid consuming hard foods for 6 months postoperatively. The patient was also advised to maintain good oral hygiene. 

He was prescribed 50 mg to 200 mg of Ketoprofen daily ad hoc, Osteogenon 840 mg 2 × 1 for 20 days (ossein–hydroxyapatite complex, Pierre Fabre Médicament Poland), and Aescin 20 mg 3 × 2 for 14 days (escin, Teva Pharmaceuticals, Warsaw, Poland). 

Patient visits were planned every two weeks during the first three months and every month until six months after the procedure. Stitches were removed 14 days after the operation. We performed two CT scans: one on the first day and the other three months after surgery. The patient underwent three months of rehabilitation of the masticatory apparatus at the local rehabilitation center. 

No complications were observed during healing. Osteosynthesis and occlusion were stable at the six-month follow-up. The healing of the patient’s mucous membrane was excellent, showing no signs of inflammation or dehiscence. The patient achieved a 3.5 cm mouth opening after four weeks and a 4.2 cm opening at the six-month mark. 

An immediate CT scan revealed the correct placement of the mandibular parts, which was consistent with the surgical plan. The linear advancement of the mandibular segment measured on the lower border of the mandible was 1.95 cm and 2.1 cm at the dental neck level. A Second CT scan showed appropriate bone formation processes at both the excision and SSO plane areas and correct mandibular segment positioning (Figure 5A,B).

At the final checkup, the patient presented with appropriate mouth opening and masticatory function, no inflammation in the surgical area, stable occlusion, and no pain. The patient was satisfied with the achieved results (Figure 5D,E).

## 3. Discussion

The secondary treatment of mandibular fractures has two main objectives: the first is to restore stable spatial relationships of the mandibular fragments and thus proper occlusion, and the second is to achieve the best possible facial aesthetics [11].

This treatment can be achieved in various ways. Referring to the described case, it might constitute a procedure consisting of microvascular free-flap transplantation at first and immediate or delayed dental implant placement [14]. This is a long procedure associated with extended convalescence and relatively high complication rates. Owing to the preparation of the donor site, the degree of surgical access must be greater, which is associated with more scarring. Additionally, the size of the transplant can sometimes lead to unaesthetic outcomes [15,16].

Otherwise, this treatment can be accomplished using the one-step approach described above. In comparison, resection combined with SSO and augmentation is a relatively short procedure. SSO is a well-established procedure in the field of maxillofacial surgery. Although its main use lies in the orthognathic field, it can be a great tool in selected trauma cases [8,9,10,11,12]. For this procedure, the placement of dental implants to restore dentition is unnecessary because of the new preplanned occlusion employing existing teeth. The presented case was planned both manually, using a classic articulator, and virtually. It incorporates novel planning software and 3-D printing procedures [17,18]. CT scans of the patient and plaster models were combined in IPS Case Designer^®^. The developed 3-D model of the mandible was transferred into the Blender^®^ software to plan the excision guides. The splints and cutting guides were relatively simple to design and manufacture. Similar planning may be performed using other types of available software. Furthermore, due to the use of an intraoral approach, scarring is minimized [13].

The main limitation of the described method is the size of the reconstruction and the possibility of complications associated with SSO [19]. There must be a bony overlap in the proximal osteotomy site; thus, the forward movement of the segment is limited. Insufficient resection of necrotic bone may lead to non-union and further loss of stabilization. Great care must be taken to avoid poor or unfavorable split lines. The risk of damaging the lower alveolar nerve posed by the presented method may also be considerably high. The regularly reported risk factors of SSO procedures such as patient’s age, smoking habits, the presence of third molars, and the type of osteosynthesis material available should always be considered when qualifying a patient for this surgery [19].

## 4. Conclusions

Although the unilateral SSO technique is well known and has been used in both reconstruction and orthognathic surgery for decades, after reviewing the accessible contemporary literature, we believe that it is the first time that it has been specifically applied to the segmentation of the mandible and the described reconstruction. It is a fairly easy, time-saving, and predictable procedure that can be incorporated into the secondary treatment of similar cases of mandibular trauma. In this case, newly achieved, stable occlusion and relatively fast recovery led to the recovery of proper masticatory function. We believe that hospitalization time and convalescence can be greatly reduced by implementing this method. Moreover, the involvement of the 3D planning and printing of patient-specific guides is worth emphasizing as it is becoming more and more widely used, reducing surgery time and costs while simultaneously increasing safety and accuracy.

## Figures and Tables

**Figure 1 reports-06-00027-f001:**
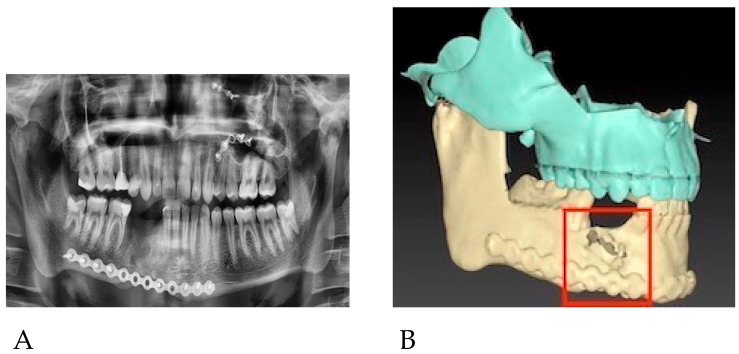
(**A**) OPG X-ray(prior to the surgery). (**B**) 3D reconstruction of the maxilla and mandible prior to surgery.

**Figure 2 reports-06-00027-f002:**
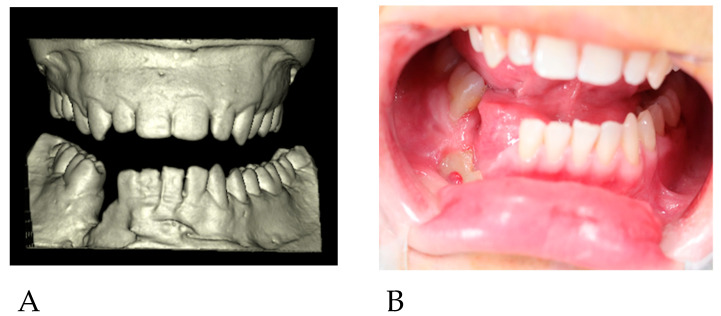
(**A**) 3D reconstruction of the plaster models. (**B**) Surgical site prior to surgery.

**Figure 3 reports-06-00027-f003:**
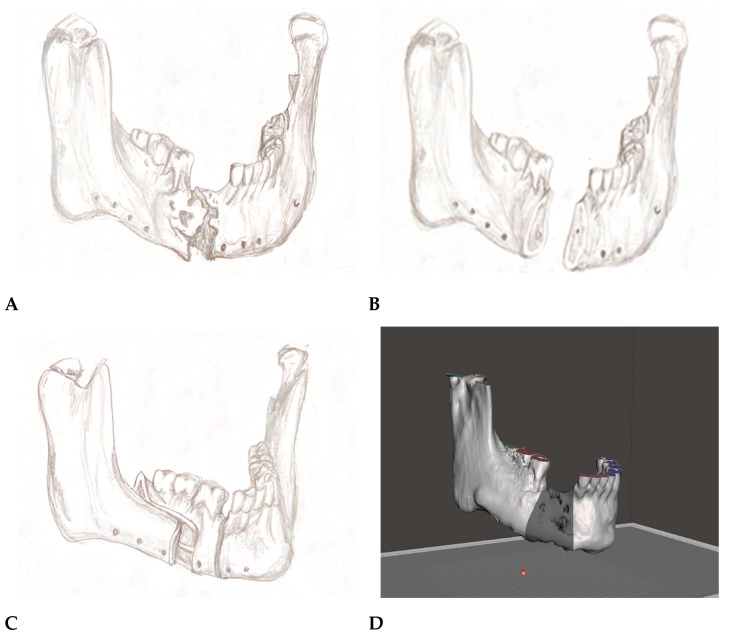
(**A**) Non-union of the bone—sketch. (**B**) Planned bone excision—sketch. (**C**) Planned segmentation and advancement—sketch. (**D**) Planned resection—3D model. (**E**) Resection splint plan—3D model. (**F**) Resection splints—3D model.

**Figure 4 reports-06-00027-f004:**
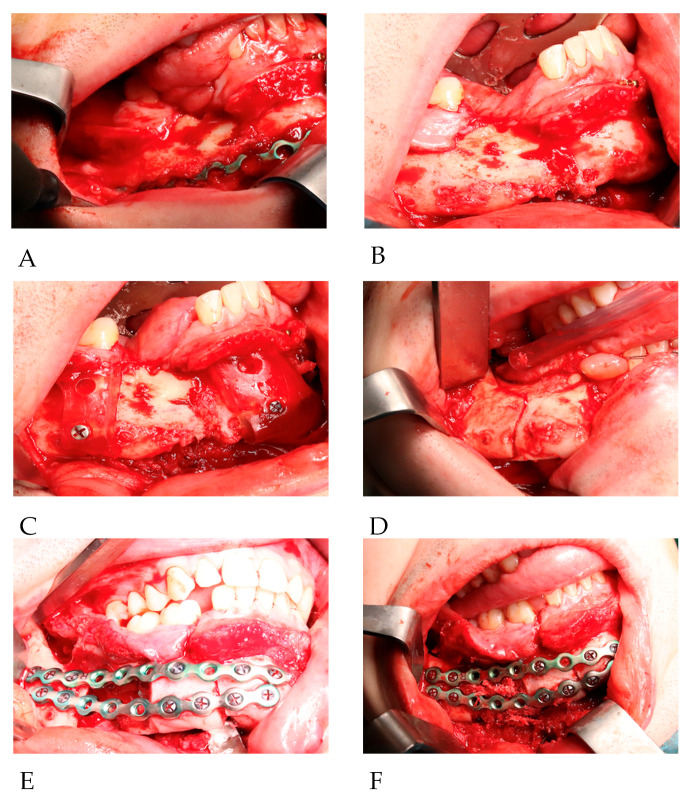
(**A**) Exposure of the reconstruction site. (**B**) Bony non-union (after plate removal). (**C**) Placement of excision splint. (**D**) SSO line before separation. (**E**) Advancement of the segment fixation in new position (final planned occlusion). (**F**) Augmentation with cancellous bone.

**Figure 5 reports-06-00027-f005:**
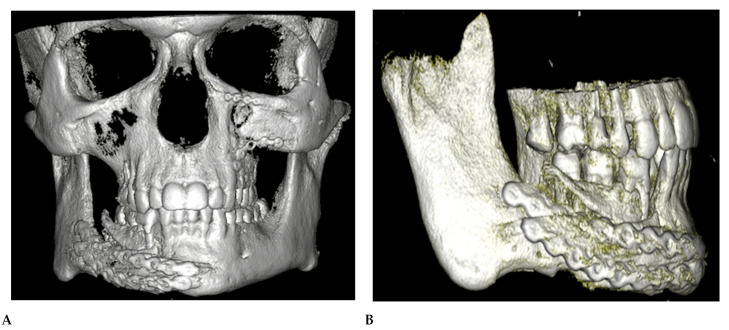
(**A**) 3D reconstruction (3 months post-op); frontal view. (**B**) 3D reconstruction (3 months post-op); lateral view. (**C**) Immediate results (post-op, intra-oral). (**D**) 2 weeks post-op, occlusion. (**E**) 2 weeks post-op, midface.

## Data Availability

All relevant data are contained within the paper.

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
