# Peer review of "Secondary Treatment of Mandibular Bone Fracture Using Sagittal Split Osteotomy and Segmentation of the Mandible: A Case Report"

_reports, 2023, doi:10.3390/reports6020027_

Round 1
Reviewer 1 Report
1. Review the format, it is not necessary to number the position of the author, it would be good to follow the template.
2. The format of the references must be numerical between square brackets.
3. the figures need description and correspondence within the text, the images are good and illustrative, however, it requires editing for size and descriptors according to text position.
4. The narrative of the case is good, the use of assisted software planning, the use of virtual and printed 3D models is a strength of the case, however, it has omissions, such as the use of acronyms such as SSO, it is used very rarely; in addition to writing errors (page 7, occlusion a d no pain. The patient was satisfied with the achieved results, it should say “and”,) these details must be taken with more attention.
5. The discussion needs more academic focus on the sso procedure; it does not establish how the combination of 3D printed models and software-assisted surgical planning were used in this case. The use of this procedure for the secondary care of mandibular traumas is novel and should highlight the results obtained as well as the follow-up of the patient.
Take attention at the format and omision of words.
Reviewer 2 Report
Dear Authors,
Please find below some observations and recommendations concerning your article entitled” Secondary treatment of mandibular bone fracture using sagittal split osteotomy and segmentation of the mandible: A case report.”
Abbreviations
- Please remove the abbreviations sections from the manuscript
In the Abstract section:
Please follow the MDPI authors' guidelines concerning the abstract structure (no more than 200 words should be included, without headings).
Acknowledgments and CRediT statement.
- Please move them at the end of the manuscript
In the Introduction section:
- Please include a paragraph on 3D planning and printing techniques in orthognatic surgery.
- Please include a paragraph about the complication associated with the mandibular sagittal split osteotomy
- Please follow the MDPI authors' guidelines:” References should be numbered in order of appearance and indicated by a numeral or numerals in square brackets—e.g., [1] or [2,3], or [4–6]”.
In the Case Report section:
2.1. Medical history.
- Please provide when did the patient first visit to the authors
- line ”A control CT scan:” please describe the first time used acronym
- Please cite all the figures in the main text as Figure 1, Figure 2.
- Please provide a description of the figures 1, 2.
2.2. Prior to secondary procedure treatment
- Please provide the presurgical details as: consent, investigations, medical approval.
- ”the patient was treated with targeted antibiotic therapy for 14 days” Please provide the class of antibiotic (product, manufacturer, City and State), dose and frequency of administration per day.
- Please cite all the figures in the main text as Figure 3, Figure 4.
- Please provide a description of the figures 3, 4.
2.3. Presurgical planning
- Please provide manufacturer, City and State of the software program, IPS Case Designer®
- Please provide product, manufacturer, City and State, material used and printing parameters of the 3D printer
- Please provide all the details about surgical acrylic splints fabrication (material, method)
- Please provide manufacturer, City and State of the software program Blender® 3D
- Please provide all the details about surgical cutting guides fabrication (material, method)
- Why didn’t you use more accurate surgical cutting guides cutting guides in conjunction with the occlusal splint guide?
- Please provide product, manufacturer, City and State of the plates and screws used.
- Please cite all the figures in the main text as Figure 5.. 10.
- Please provide a description of the figures 5-10.
2.4. Surgical procedure
- Please provide more information regarding if the inferior alveolar nerve was preserved.
- Please cite all the figures in the main text as Figure 11.. 16.
- Please provide a description of the figures 11-16.
- Please provide the postoperative treatment and indications given to the patient.
- Please remove the” 3. Results” or replace it with ”2.5 The outcomes” subchapter.
- Please rearrange the figures without overlapping them.
- Please cite all the figures in the main text as Figure 17.. 21.
- Please provide a description of the figures 17-21.
In the References section: please follow the styles recommended for MDPI journals.
Reviewer 3 Report
Authors incorporated 3D planning and printing procedures, with segmentation of the mandible with unilateral sagittal split osteotomy and vertical osteotomy of the mandibular body surgical techniques for mandibular bone fractures.
The content in introduction can not support the research gap of this study. I can not get any idea of the significance of this study design. From the results, this novel technique may seem to be positive, such as good recovery. But, the problem is that I can not have a clear clue regarding to the research gap and significance in this study. No relevant comparison study was offered. So, I am concern about the significance of this study design.
Minor editing of English language required
Round 2
Reviewer 1 Report
Consider including Author Contributions information acording to
CRediT taxonomy
Improve compare to old version.
Reviewer 2 Report
Dear authors,
Thank you very much for revising the manuscript according to my comments.
Reviewer 3 Report
Agree for publication
minor modification